# Effect of Residual Oxygen Concentration on the Lattice Parameters of Aluminum Nitride Powder Prepared via Carbothermal Reduction Nitridation Reaction

**DOI:** 10.3390/ma15248926

**Published:** 2022-12-14

**Authors:** Jaegyeom Kim, Heewon Ahn, Seung-Joo Kim, Jong-Young Kim, Jae-Hwan Pee

**Affiliations:** 1Icheon Branch, Korea Institute of Ceramic Engineering and Technology (KICET), 3321, Gyeongchung Rd., Sindun-Myeon, Icheon-si 17303, Republic of Korea; 2Department of Chemistry, Ajou University, Suwon 16499, Republic of Korea; 3Department of Energy Systems Research, Ajou University, Suwon 16499, Republic of Korea

**Keywords:** aluminum nitride, residual oxygen, lattice parameter, nitrides

## Abstract

Residual oxygen in wurtzite-type aluminum nitride (AlN) crystal, which significantly affects phonon transport and crystal growth, is crucial to thermal conductivity and the crystal quality of AlN ceramics. In this study, the effect of residual oxygen on the lattice of AlN was examined for as-synthesized and sintered samples. By controlling reaction time in the carbothermal reduction nitridation (CRN) procedure, AlN powder was successfully synthesized, and the amount of residual oxygen was systematically controlled. The evolution of lattice parameters of AlN with respect to oxygen conc. was carefully investigated via X-ray diffraction analysis. With increasing amounts of residual oxygen in the as-synthesized AlN, lattice expansion in the *ab* plane was induced without a significant change in the *c*-axis lattice parameter. The lattice expansion in the *ab* plane owing to the residual oxygen was also confirmed with high-resolution transmission electron microscopy, in contrast to the invariant lattice parameter of the sintered AlN phase. Micro-strain values from XRD peak broadening confirm that stress, induced by residual oxygen, expands the AlN lattice. In this work, the lattice expansion of AlN with increasing residual oxygen was elucidated via X-ray diffraction and HR-TEM, which is useful to estimate and control the lattice oxygen in AlN ceramics.

## 1. Introduction

In recent years, aluminum nitride (AlN) has attracted considerable attention in the electronics industry because of its excellent thermal and electrical properties, such as high thermal conductivity (~320 W/mK), excellent insulating performance, thermal expansion coefficient close to that of silicon, and low dielectric constant [1,2,3]. Therefore, AlN has been used in various applications, such as in heat-dissipative materials in power electronic devices and heater/electrostatic chucking dielectrics in semiconductor manufacturing equipment. The carbothermal reduction nitridation (CRN) process is mainly used in the industrial synthesis of high-purity AlN powder [4,5]. To manufacture high-quality AlN ceramics, high-purity AlN powder material with low O/C conc. is required. Several studies have shown that oxygen dissolution in the AlN lattice during synthesis has a decisive effect on the quality of AlN ceramics, determining thermal conductivity and defect density [6,7,8,9,10,11].

However, the relationship between the lattice parameter and the residual oxygen of AlN found in previous works on sintered specimens is contradictory to works on epitaxial film on substrates [6,9,10]. G.A. Slack reported that the *c*-axis lattice parameter decreases from 4.981 to 4.978 Å as residual oxygen increases [6]. On the other hand, in wurtzite structures such as AlN and GaN, the lattice volume is found to be increased in theoretical and experimental ways when nitrogen is replaced by oxygen [12,13,14]. For sintered AlN [9,10,11], the correlation between the lattice parameter and the residual oxygen might be obscured by oxygen gettering due to the sintering agents of the oxides. The driving force behind lattice expansion is to relax the increased total energy of doped crystal. When impurities of oxygen or silicon are incorporated into the crystal, electrons in the conduction band should increase the total energy of the crystal; hence, to decrease the energy, the level of the conduction band is lowered, which is related to a change in lattice volume. Therefore, a more reliable and precise estimation of oxygen impurity in AlN lattices is required to enhance the quality of AlN ceramics.

In this study, we synthesized AlN powder with controlled oxygen concentration in the lattice and systematically investigated the evolution of the lattice parameters of AlN via X-ray diffraction (XRD). The structural changes with respect to residual oxygen were verified with transmission electron microscopy (TEM). Based on these results, the correlation between oxygen concentration and the changes in AlN lattice parameters is discussed.

## 2. Materials and Methods

High-purity α-Al_2_O_3_ powder and carbon black powder were purchased from Chalco Shandong Co., Ltd. and Columbian Chemicals Company, respectively, for the synthesis of AlN powder. Raw materials of Al_2_O_3_ and carbon black were uniformly mixed in a molar ratio of 1:3.1 by wet ball milling using anhydrous ethanol. The resultant slurry of Al_2_O_3_ and carbon black was then completely dried in a drying oven at 150 °C. The powder mixture (~20 g) was placed into a graphite crucible and then heated at 1700 °C for 0.5 to 6 h in an electrical furnace. The CRN process was performed under high-purity N_2_ gas with a flow rate of 3 L/min. Subsequently, powder samples were heated in air at 700 °C for 1 h to remove the remaining carbon.

Phase identification and lattice parameter measurement were carried out using an X-ray diffractometer (D/max 2500 v/pc, Rigaku, Japan). Data were collected in the 2-theta range of 20–80° with a step size of 0.02° and a counting time of 1 s for each step at room temperature. The lattice parameters were determined using the Le Bail method by mixing Si powder (Alfa Aesar, 99.5%) as an internal standard. The lattice parameter of the Si phase was fixed at *a* = 5.4309 Å, and only the lattice constants *a* and *c* of the hexagonal AlN phase were refined as variables. The oxygen content of the AlN powder was measured using an oxygen/nitrogen analyzer (EMGA-920, HORIBA, Kyoto, Japan). More than three samples were tested to obtain the average value. The microstructure was observed with high-resolution transmission electron microscopy (HRTEM; ARM-200F, JEOL, Tokyo, Japan).

## 3. Results

### 3.1. Synthesis

To investigate the effect of CRN reaction time on nitridation rate, the powder mixture of alumina and carbon was reacted at 1700 °C for 0.5–6 h. XRD patterns of the AlN powder samples after the CRN reaction are shown in Figure 1. X-ray diffraction peaks of all the samples that were reacted for 1 h or more were found to correspond to monophasic AlN with a wurtzite structure (JCPDS no. 25-1133). In contrast, a small amount of Al_2_O_3_ remained in the sample reacted for 0.5 h due to the incomplete conversion of α-Al_2_O_3_ to AlN. This indicates that a CRN reaction time of at least 1 h (@1700 °C) is required to convert α-Al_2_O_3_ into the wurtzite AlN phase completely.

Lefort et al. suggested the mechanism of CRN reaction involves the dissociation of alumina controlled by very low partial pressures of oxygen in contact with carbon, followed by the quick nitridation of the formed metallic vapor [15]. At temperatures of 1600 °C or higher, the CRN reaction rate increases rapidly at the initial stage but gradually decreases after 1 h. Table 1 shows the change in oxygen content in the AlN samples according to the CRN reaction time at 1700 °C. As the CRN reaction time increases, the residual oxygen content gradually decreases, indicating that a reaction time of at least 5 h is required for the completion of the CRN process at 1700 °C.

According to ^27^Al solid-state NMR analysis, the majority of the residual oxygen is found to be dissolved in the AlN lattice as an AlN_3_O defect with tetrahedral symmetry. Detailed analysis results and the method are presented in the Appendix A.

### 3.2. Oxygen Dissolution Effect on AlN Lattice

#### 3.2.1. X-ray Diffraction

Lefort et al. suggested the mechanism of CRN reaction involves the dissociation of alumina controlled by very low partial pressures of oxygen in contact with carbon, followed by quick nitridation of the metallic vapor [15]. At temperatures of 1600 °C or higher, the CRN reaction rate increases rapidly at the initial stage but gradually decreases after 1 h. Table 1 shows the change in oxygen content in the AlN samples according to the CRN reaction time at 1700 °C. As the CRN reaction time increases, the residual oxygen content gradually decreases, indicating that a reaction time of at least 5 h is required for the completion of the CRN process at 1700 °C.

At first sight, the lattice parameter of AlN is expected to decrease with the increase in oxygen content because of the smaller tetrahedral sharing radius of oxygen than that of nitrogen and the occurrence of metal vacancy at the aluminum site [6]. According to previous reports, the relationship between the lattice constant and residual oxygen of AlN is found to be quite confusing [6,9,10]. G.A. Slack reported that with increasing residual oxygen, the *c*-axis lattice constant decreases from 4.981 to 4.978 Å [6]. J. Harris et al. reported that unit cell volume increases by 0.1% until oxygen concentration reaches 0.75 at%, and then the volume decreases by 0.06% from 0.75 to 3.0 at% [9]. Previous studies focused on the relationship between oxygen content and physical properties such as thermal conductivity, and therefore lattice oxygen concentration was investigated on sintered ceramics or single crystals. Ceramic samples of AlN were prepared either by hot-press sintering without sintering aids at >1900 °C [6], or pressure-less sintering with aids [16,17,18]. The majority of studies on ceramics use sintering aids such as yttrium oxide, whereas single crystalline samples with very low oxygen dissolution are also reported in the literature [6,19,20].

On the other hand, theoretical and experimental works on O-doped GaN crystals and epitxial films evidence lattice expansion by size, electronic effects [12,13,14], and strains on the AlN lattice [21]. In this work, the measurement was made on as-synthesized powder with a carbothermal reaction at 1700 °C using high-purity alumina powder with carbon. Therefore, a major difference between previous reports and our work is the local symmetry of oxygen in the AlN material. In our nitridation reaction, the oxygen of alumina is removed by the formation of volatile carbon dioxide, and the oxygen remains as a form of AlN_3_O in the AlN lattice with a wurtzite structure, which is shown by ^27^Al solid-state NMR spectra, showing the shoulder AlN_3_O peak beside the AlN_4_ main peak (Appendix A). O atoms are distributed in the parent AlN_4_ lattice with random periodicity, possibly with a complex consisting of O_N_ + V_Al_/e^−^. Meanwhile, dissolved oxygen turns into Al-O-N phase or second phases (YAG,YAP) after sintering, which is shown by ^27^Al solid-state NMR analysis (Appendix A). The Al-O-N phase was not detected in our powder XRD analysis; however, the presence of the Al-O-N defect with both octahedral and tetrahedral symmetry was evidenced by the Al-NMR spectra. Therefore, the parent AlN lattice should be influenced by the sintering reaction because the amount of oxygen dissolved in the parent AlN lattice changed. The lattice parameters of parent AlN might be influenced by the formation of the Al-O-N phase with short range order and/or second phases.

In this work, XRD diffraction results clearly show that increased residual oxygen induces the expansion of the *a*-axis lattice parameter with the *c*-axis lattice parameter almost intact. The lattice constants for our AlN materials were determined using the Le Bail method by mixing Si powder as an internal standard. A manually selected background was used, and the peak profile was fitted using the Thompson–Cox–Hastings pseudo-Voigt function of the FullProf program [22,23]. Figure 2 shows the changes in the lattice parameters of AlN as a function of the CRN reaction time. The *a*-axis lattice constant decreases gradually from 3.11286(2) to 3.11140(2) Å as the CRN reaction time increases. In contrast, the *c*-axis lattice parameter remains almost constant at ~4.9807 Å regardless of reaction time. The unit cell volume gradually decreases as the reaction time increases, and almost no change was observed after 5 h.

On the other hand, the lattice parameters for all the sintered samples are *a* ~ 3.1115 Å, *c* ~ 4.9805 Å, which are effectively the same as those of single crystalline AlN by Schulz et al. (*a* = 3.110 Å, *c* = 4.980 Å) and by Kumagai et al. (*a* = 3.1111 Å, *c* = 4.9808 Å) [19,20]. This might be due to oxygen gettering from the AlN grains by the sintering aid, which leads to the significant removal of lattice oxygen, except for the residual Al-O-N and second phases (YAG,YAP), as shown by the Al-NMR spectra for the sintered samples (Appendix A).

#### 3.2.2. HRTEM Analysis

The changes in the AlN lattice due to oxygen defects were also investigated using HRTEM. TEM images of the reaction samples for 1 h (denoted as 1 h) and 5 h (denoted as 5 h) with apparent differences in lattice parameters are shown in Figure 3 and Appendix A. The AlN samples were measured along the [001] zone axis to confirm the interplanar distance between the (100) and (010) planes (Figure 3a,c). The distance between the (100) planes for the 1 h sample was 2.750 Å, but 2.648 Å in the 5 h sample, confirming that the lattice of the 5 h sample was contracted more along the *a*-axis than that of the 1 h sample. In contrast, the interplanar distance between the (002) planes were ~2.50 Å in both the samples. The interplanar distances observed in the TEM measurements are consistent with the XRD result; as the amount of residual oxygen content decreases, the unit cell volume shrinks owing to the decreased *a*-axis lattice parameter. This implies that the oxygen-induced defects in the AlN lattice cause deformation in the *ab* plane, whereas the deformation in the *c*-axis direction is limited to a negligible magnitude.

### 3.3. Micro-Strain Analysis

Oxygen impurity also generates a strain on AlN crystals, which is reflected in the powder X-ray diffraction profiles. In general, crystal imperfection induced by the formation of a vacancy or the inclusion of an impurity ion into the lattice may cause a non-uniform strain (micro-strain) on the materials, leading to peak broadening in the XRD pattern [24]. To estimate the micro-strain in the AlN samples synthesized, Williamson–Hall (W-H) analysis was performed, and the micro-strain was deduced from the slope of the plot of βcosθ/λ versus 4sinθ/λ [25].

Figure 4 shows the correlation between the micro-strain in the AlN lattices (a detailed W-H plot is shown in Appendix A). The micro-strain gradually decreases from 4.7(1) × 10^–6^ to 3.9(2) × 10^–6^ as the reaction time increases, owing to the decrease in the residual oxygen concentration. The strain was saturated at 3.9(2) × 10^–6^ after 5 h, as were the changes in the residual oxygen concentration and lattice constant. This suggests that the micro-strain of AlN decreases along with the decrease in defect concentration.

## 4. Discussion

In this work, the evolution of lattice parameters with respect to oxygen concentration for as-synthesized AlN differs from that of the sintered AlN. The oxygen is dissolved in the form of 4-coordinated AlN_3_O in a wurtzite-type AlN lattice for the as-synthesized samples. Therefore, the lattice parameters of the as-synthesized AlN represent, in fact, those of O-doped AlN, which can be explained by DX center formation [26]. According to first-principle calculation, the O atom is situated on the substitutional N site with a 4% elongation of Al-O bond length compared with 1.90 Å Al-N bond length in the positively charged state of O_N_^+^, whereas in the O_N_^−^ state, the O atom is displaced toward the third nearest neighbor of Al by 0.9 Å in the [0001] direction because of Coulombic attraction, and Al-O bond length is reduced to 2.06 Å in the wurtzite AlN structure. The combination of these two effects would lead to an increased *a*-axis lattice parameter with, in effect, the same *c*-axis lattice parameter.

Meanwhile, for the sintered samples, the 4-coordinated defect turns into a 6-coordinated Al(N,O)_6_ defect in the form of Al-O-N, or the well-known second phases of YAG,YAP with the formation of aluminum vacancy (V_Al_). Therefore, the lattice parameters of the sintered samples represent those of the AlN lattice, in which residual oxygen is gettered by reaction with sintering aids such as Y_2_O_3_. In our work, the lattice parameters of the sintered samples are almost the same as those of single crystalline AlN, as stated in the Results section. This result implies that most of the lattice oxygen is removed or transformed into second phases or Al-O-N, which is also supported by the Al NMR analysis results.

For GaN with the same wurtzite structure as AlN, the lattice volume increases when nitrogen is replaced by oxygen [12,13,14]. According to van de Walle, the size effect(Δ*a*/*a*) due to oxygen substitution is calculated to be 1.0 × 10^−24^ cm^3^ [O_defect_] ([O_defect_] ~ 10^21^/cm^3^ in our case) and then Δ*a*/*a* ~ 0.1% for our AlN [26]. The experimental change in the *a*-axis lattice constant is found to be at most 0.06%, and that of the *c*-axis parameter remains the same. In the wurtzite structure (e.g., AlN, GaN), the displacement of oxygen due to coulombic attraction between O_N_^−^ and Al^3+^ compensates lattice expansion along the *c*-axis, which makes the *c*-axis lattice parameter effectively the same. Such an invariant *c*-axis lattice parameter was also reported in doped wurtzite-type AlN crystals [27,28]. These results indicate that the structure of AlN with oxygen-induced defects would be stabilized by inducing an expansion in the *ab* plane rather than along the *c*-axis.

## 5. Conclusions

Wurtzite-type AlN powder with different levels of residual oxygen was synthesized using the CRN reaction, and it was confirmed that Al_2_O_3_ raw materials were converted to the AlN phase after 1 h at 1700 °C. Most of the residual oxygen was found to be dissolved in the lattice by ^27^Al solid-state NMR analysis, which diminished as the reaction time increased. For the as-synthesized AlN, the increased *a*-axis lattice parameter and the expansion of lattice volume owing to the oxygen-induced defects in the AlN lattice were confirmed by XRD and HRTEM analyses. The lattice expansion behavior due to the oxygen dissolution of the as-synthesized AlN was in contrast with the previously reported lattice shrinkage along the *c*-axis of the sintered AlN, which can be attributed to the different chemical environment around the residual oxygen. The oxygen is dissolved in the parent AlN lattice with tetrahedral coordination for the as-synthesized material; on the other hand, the oxygen is present in the Al-O-N and second phases (YAG,YAP) with octahedral coordination for the sintered samples. Moreover, XRD peak broadening with the W-H plot confirmed that lattice volume expansion occurs as a result of the stress induced by residual oxygen. This study enables us to understand the effects of residual oxygen in AlN on long-range structures and the local strain on Al-N phonons.

## Figures and Tables

**Figure 1 materials-15-08926-f001:**
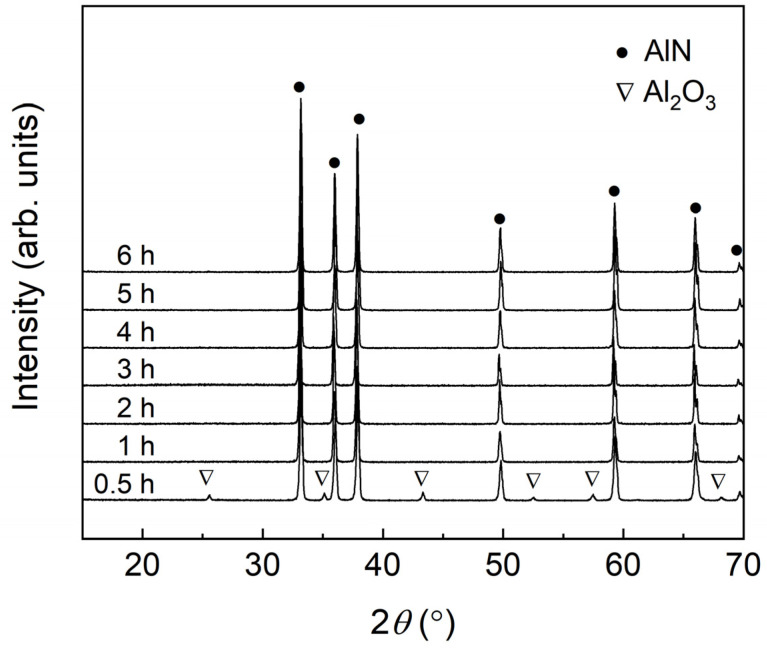
XRD patterns of the products obtained at different reaction times at 1700 °C.

**Figure 2 materials-15-08926-f002:**
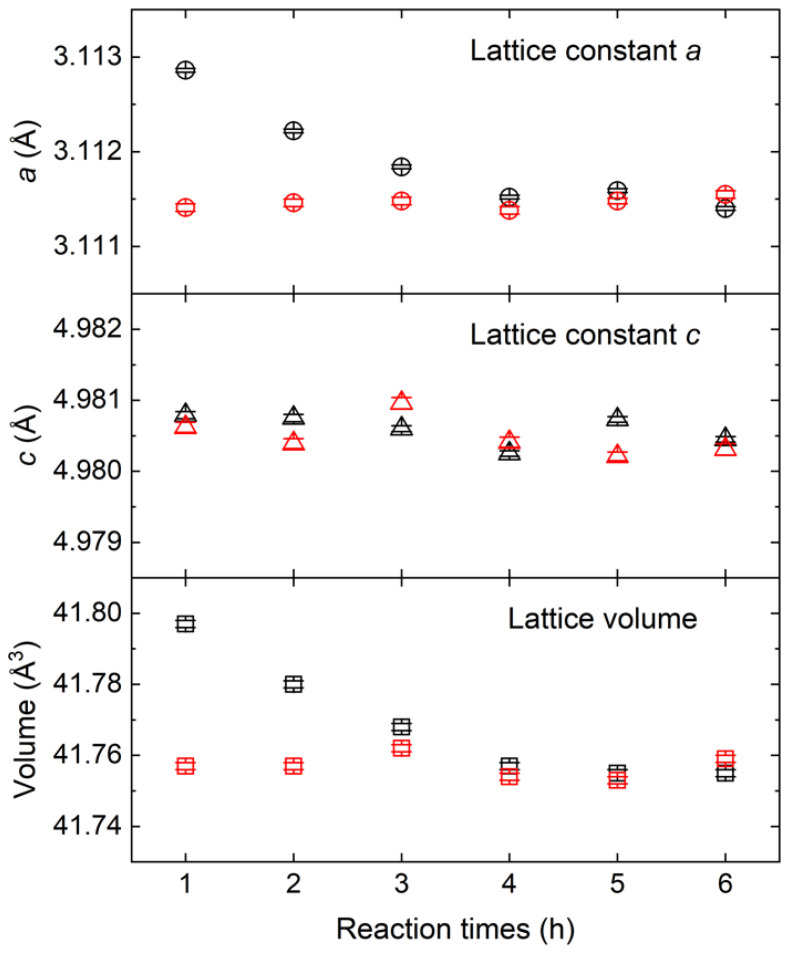
Changes in lattice parameters (*a*, *c*) and unit cell volume of AlN samples with the reaction time. The lattice constants of the AlN samples before and after sintering are shown in black and red open symbols, respectively.

**Figure 3 materials-15-08926-f003:**
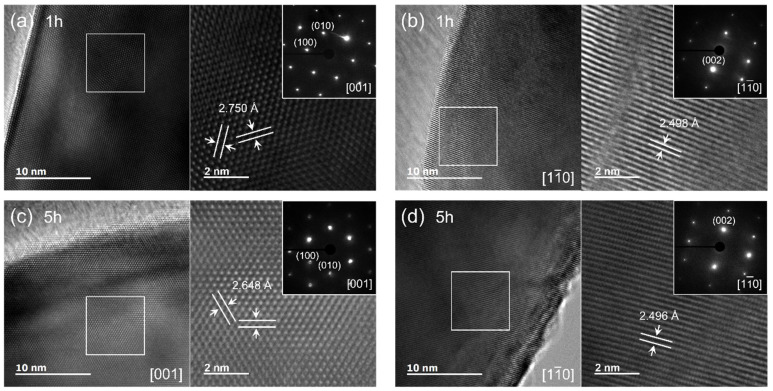
HRTEM images of 1 h (**a**,**b**) and 5 h (**c**,**d**) samples corresponding to the electron diffraction patterns shown as insets.

**Figure 4 materials-15-08926-f004:**
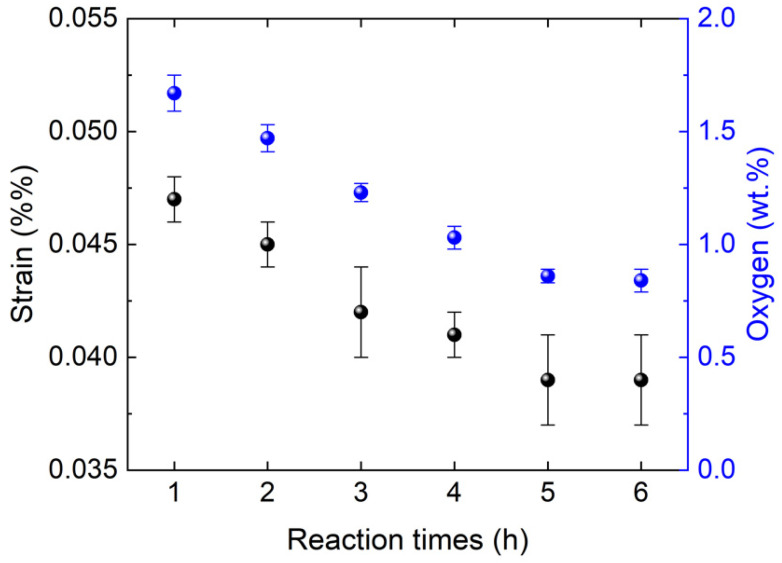
Changes in strain and residual oxygen concentration AlN lattice with reaction time.

**Table 1 materials-15-08926-t001:** Oxygen concentration of AlN samples after decarbonization.

Reaction Time (h)	O (wt%/at%) *	C (ppm) *	AlN_3_O NMR Peak Area (%) **
1	1.67(8)/4.28(21)	-	-
2	1.47(6)/3.77(15)	1691(16)	11.37
3	1.23(4)/3.15(10)	1527(14)	8.63
4	1.03(5)/2.64(13)	1412(13)	7.17
5	0.86(3)/2.20(8)	1087(10)	4.72
6	0.84(5)/2.15(13)	1132(10)	4.75
Commercial	0.85(5)/2.17(13)	-	0

* Standard deviation values of remaining O and C conc. are presented in parentheses. ** The peak area means the proportion of the area, due to AlN_3_O defects, to the total area of main peak at ~120 ppm.

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
