# Peer review of "Effect of Residual Oxygen Concentration on the Lattice Parameters of Aluminum Nitride Powder Prepared via Carbothermal Reduction Nitridation Reaction"

_materials, 2022, doi:10.3390/ma15248926_

Round 1
Reviewer 1 Report
Due to their thermal conductivity, sintered ALN ceramics are of great importance for electronic devices and studies on the underlying mechanisms are worth publishing. The relationship between lattice parameters and oxygen content, which was addressed in this work, plays an important role in this context, as oxygen in the lattice degrades the thermal properties.
However, such studies have been available for a long time, older ones are also cited here (e.g. Ref. [6] from 1973), but there are clearly more recent studies available (e.g. Piero Gasparotto, ACS Appl. Mater. Interfaces 2021, 13, 5762-5771, where even theory and experiment is compared). These studies have in common that they resulted in an accepted dependence of the lattice parameters on the oxygen content, which is now even used to calibrate the oxygen content in AlN ceramics via X-ray diffraction (see e.g. V. V. Zakorzhevskii, Inorganic Materials, 2021, Vol. 57, No. 10, pp. 998-1004, where several papers on this were used as a reference).
It is therefore all the more surprising to read here about a deviating dependence. The data seem to be consistent, the only noticeable thing for me is that at higher angles 2Theta in image 1 shoulders or even side peaks seem to appear (especially at 10.3 and 11.0 reflections). This lacks a discussion of what these are and how they were accounted for.
The remarkable differences from the literature data should definitely be highlighted more clearly in this paper and, if possible, their causes should also be discussed, both why the material behaves differently (due to specifics of the synthesis?) and what mechanisms might be behind it. The explanation about the DX centre does not seem very convincing. The strong displacement of the O atom would practically create a new Al-O bond in the c-direction, while the old one would be broken. This would also shift the new binding partner for O in the c-direction towards the O atom, which would correspond to a lattice compression in the c-direction. However, this contradicts the experimental results of this work.
Therefore, as it stands, I would not recommend publication.
Some further points:
The use of the term lattice constant is becoming increasingly uncommon, with lattice parameter being used instead (see also the headings of references 12, 13).
The references are almost all more than 10 years old, many more than 20 years old, although there are quite a few recent papers. Here an update might be useful.
The first section of 3.2.1 repeats the second of 3.
Author Response
Please see attachement.

Reviewer 2 Report
The manuscript is well-written, clear, and easy to understand. The novelty of the work is doubtful, but the investigation of residual oxygen concentration on the lattice constants of aluminum nitride powder is done systematically and concisely. Please, correct the notation of the Figures in Supplementary Material (two Figures have the same S2 notation).
Reviewer 3 Report
1. In table 1, there are three experimental data such as 1.67(8)/4.28 in the second column O (wt%/at%)), I suggest you explain what the data in the brackets. Also please carefully explain the meaning of C ppm and NMR peak area%.
2. According to 27Al Solid-state NMR analysis, majority of the residual oxygen is found to be dissolved in the AlN lattice as AlN3O defect having tetrahedral symmetry. So please consider put the Al NMR analysis in the main manuscript.
3. This work is of scientific importance. However, it does not appear to be sufficiently well written. There are some language, grammar, reference citation and unit errors. For example, he micro-strain gradually decreased decreases from 4.7(1) × 10–6 to 3.9(2) × 10–6 as the reaction time increases owing to the decrease of the residual oxygen concentration.
4. There is some language, grammar, reference citation and unit errors. Please polish the whole manuscript.
a. Please add some comments or footnote for Table 1.
b. Please add indent at the beginning of all paragraphs.
c. Please put a space before the unit.
d. Please check all the manuscript and modified detailed problem. Such as “The micro-strain gradually decreased decreases from 4.7(1) × 10–6 to 3.9(2) × 10–6 as the reaction time increases owing to the decrease of the residual oxygen concentration.”
e. Please put the reference citation number before the full stop.
5. In Table 1, there are three experimental data such as 1.67(8)/4.28 in the second column O (wt%/at%)). The data in this table is confusing. I suggest you explain what the data in the brackets. Also please carefully explain the meaning of C ppm and NMR peak area%.
6. According to 27Al Solid-state NMR analysis, majority of the residual oxygen is found to be dissolved in the AlN lattice as AlN3O defect having tetrahedral symmetry. I think the NMR analysis is very important for your manuscript. So please consider put the Al NMR analysis in the main manuscript.
7. What’s the difference of your study on the detecting the lattice oxygen in ceramics? Please emphasize the novelty of the paper.
Author Response
Please see attachment.
p.s. We will get English correction service from MDPI.

Round 2
Reviewer 3 Report
Further revision has been made with great improvement according to the comments from reviewers. I suggest the revised manuscript can be accepted.